# Revisiting Incremental Object Detection with Pre-Trained Vision-Language Models

## Abstract

Pre-trained Vision-Language Models (VLMs) have recently been applied to Incremental Object Detection (IOD), achieving notable progress. However, existing researches often oversimplify real-world scenarios by assuming the incremental tasks come from a single general domain. To better investigate VLMs under IOD, it is necessary to explore more generalized scenarios that encompass both novel categories and domains. To this end, we propose Cross-Domain Incremental Object Detection (CDIOD), a new benchmark that assesses the ability to continuously adapt to diverse object detection tasks across domains. CDIOD reveals that existing methods struggle to balance between adaptivity and stability under substantial domain shifts. To tackle this challenge, we propose $D^3$, a novel framework that possesses **D**ynamic grouping to promote knowledge sharing and prevent task collisions; **D**ynamic adapter assignment to effectively adapt to new tasks while controlling model scale; and **D**ynamic training pipeline to ensure a proper stability-adaptivity balance. $D^3$ enables VLMs to effectively handle task streams of various distribution shifts. Extensive experiments demonstrate that $D^3$ achieves state-of-the-art results across three benchmarks, highlighting its versatility and robustness in diverse incremental learning scenarios.

## 1 Introduction

Object detection is a fundamental task in computer vision, with broad applications in areas such as autonomous driving [70; 5], remote sensing [31; 35], and underwater object detection [67; 23]. Traditional detectors are constrained to specific domains and pre-defined categories [52; 3], requiring costly retraining for each new scenario. This limits their adaptivity in dynamic real-world environments. Recent advances in Vision-Language Models(VLMs) [32; 75; 39] offer an innovative object detection paradigm. After pre-trained on large-scale image-text pairs [21; 50], VLMs can recognize diverse visual concepts, enabling a single model to perform various detection tasks without retraining [53; 55]. Despite strong zero-shot capability, deploying these models for diverse, specialized downstream tasks (e.g., remote sensing, underwater) often requires fine-tuning to bridge the distribution gap. In practice, when data arrives non-stationarily, continuously finetuning models often leads to catastrophic forgetting of previously learned knowledge and decrease of zero-shot capabilities. [74].

Incremental learning is essential for VLMs to remain adaptable in dynamic environments while mitigating catastrophic forgetting. However, most existing IOD researches [41; 73; 26; 63] oversimplify real-world challenges, assuming incremental learning occurs within a single, general domain. This consumption conflicts with the design of modern VLMs [32; 39], which are fundamentally intended to operate across diverse domains. Under this simplified setting, naively fine-tuning pre-trained VLMs already matches SOTA as shown in Fig. 1(b). This highlights that existing IOD benchmarks can no longer adequately reflect the incremental learning capabilities of modern VLM-based detectors and overlook the challenges in real-world scenarios. In practice, evolving domains and novel categories often coincide, posing compounded challenges. A comprehensive evaluation protocol is essential to assess VLMs' capabilities and reflect real-world challenges in a realistic way.

To better understand the challenges of incremental learning with VLMs, we introduce a new challenging yet practical benchmark named Cross-Domain Incremental Object Detection (CDIOD). As shown in Fig. 1(a), CDIOD comprises datasets from three different domains: remote sensing(DIOR),

natural scenes (Pascal VOC) and underwater (RUOD). Each dataset is further divided into several sub-tasks. Models are required to incrementally learn all these tasks and are finally evaluated on three datasets in a task-agnostic manner. We compare several SOTA methods under this more generalized setting. The results show that VLMs suffer from severe forgetting when exposed to substantial domain shifts from their pre-training distribution (e.g., from Objects365 to remote sensing). The forgetting is further exacerbated by the need of handling novel categories. This compounded challenge leads to a dilemma, making it difficult to pursue an optimal subspace for previous and upcoming knowledge. This consequently makes existing methods fail to effectively balance between stability and adaptivity. As shown in Fig. 2, full fine-tuning approaches like GCD [63] adapt well to new domains but fail to retain prior knowledge under large distribution shifts. In contrast, PEFT-based methods such as MD-DETR [2] and Zira [6] better preserve pre-trained knowledge but exhibit limited adaptivity. This underscores that incremental learning of VLMs under diverse and evolving scenarios remains an underexplored problem.

To address these challenges, we propose $D^3$, a novel framework that jointly enhances adaptivity, stability, and efficiency. $D^3$ begins with Dynamic Task Grouping (DTG), which groups tasks by distribution similarity to guide the entire learning process. Building on this, Dynamic Adapters Assignment (DAA) allocates and manages Incremental Group Adapters (IGA) at group-level, enabling knowledge sharing across related tasks while isolating unrelated ones. The dynamic training pipeline then adjusts learning strategies based on grouping: new groups initialize adapters to tackle large distribution shifts, while existing groups consolidate adapters to maintain stability and encourage knowledge transfer with controlled parameter growth. At inference, $D^3$ performs group-wise routing for each input, significantly reducing routing errors. This dynamic design enables VLMs to learn from evolving task streams with minimal forgetting and parameter overhead.

Extensive experiments on three benchmarks Fig. 1(b), including CDIOD, conventional IOD, and IVLOD[6], demonstrate that $D^3$ consistently balances adaptivity and stability across diverse incremental settings. It achieves competitive performance with significantly fewer parameters, and its modular design enables scalable, efficient adaptation to a wide range of downstream detection tasks.

**Our contributions are threefold:**

- We introduce a novel benchmark, CDIOD, to evaluate the capability of VLMs in more generalized scenarios that involve substantial domain shifts, which existing IOD methods struggle to handle.
- We propose $D^3$, a dynamic framework that integrates task grouping, adaptive adapter assignment, and knowledge consolidation to jointly enhance adaptivity, stability, and efficiency, achieving a 16.5 AP gain on CDIOD with only 1.2% additional parameters.
- Extensive evaluations across three benchmarks confirm our method's consistent SOTA performance, validating its generality and robustness in diverse incremental settings.

## 2 RELATED WORKS

### 2.1 INCREMENTAL LEARNING

Incremental learning aims to enable models to continuously learn new tasks without forgetting previous knowledge [43; 60; 13]. Existing approaches can be broadly categorized into three paradigms. Regularization-based methods impose constraints on the model to prevent overfitting to new data, either through explicit penalties on model weights [27; 29; 72] or implicit constraints via knowledge distillation (KD) [16; 36; 59; 66]. Rehearsal methods maintain a memory buffer of previously seen images [51; 48] or intermediate features [45; 20], and selectively replay them during subsequent incremental stages [17]. Architectural methods dynamically expand network [69; 34; 68] to accommodate new knowledge without interfering with existing ones.

### 2.2 INCREMENTAL OBJECT DETECTION

Compared to incremental classification, IOD is more challenging due to the presence of both old and new classes in the same image, leading to missing annotations and background shift. Some works extended aforementioned methods to object detection. For KD-based approaches [57; 40; 46; 47; 10; 24; 63], RILOD [57] first applied LwF [36] to IOD. ERD [10] further filters negative responses. For rehearsal methods [1; 41; 42; 25], CL-DETR [41] replays exemplars that aligned with

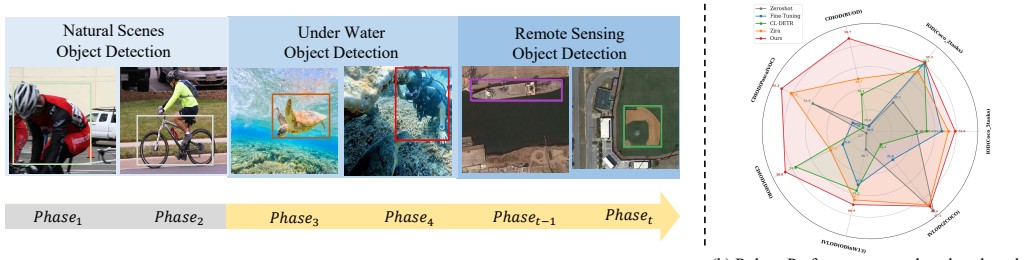

(a) Cross-Domain Incremental Object Detection        (b) Robust Performance over three benchmarks

Figure 1: (a) Illustration of incremental learning spanning across different domains (Natural Scenes, Underwater, Remote Sensing). A single-domain process (e.g., grey area) corresponds to conventional IOD, whereas the full sequence defines CDIOD. (b) Radar chart comparing methods across various incremental scenarios (e.g., CDIOD, IOD(COCO), IVLOD(OdinW13)). The outer boundary indicates joint training performance, except for IVLOD(ZCOCO), where zero-shot serves as the upper bound.

training data distribution, and ABR [42] replays only foreground objects to mitigate foreground shift. Leveraging VLMs, [73] assign separate parameters for each task to reduce interference. GCD [63] distills vision-language topological relationships to preserve semantic structures. Zira [6] introduces reparameterizable modules to adapt while retaining pre-trained knowledge.

### 2.3 Parameter-Efficient Fine-tuning (PEFT)

PEFT techniques adapt pre-trained models to downstream tasks by updating a small subset of parameters, significantly reducing computational costs. Prompt tuning methods [33; 22] learns task-specific prompts to guide model predictions. Adapter-based methods [18; 15] insert trainable bottleneck modules into each transformer layer. LoRA [19] approximates weight updates via low-rank matrices. These techniques have been extended to incremental learning. For example, [65; 64; 58; 28] use learnable prompts to encode task-specific knowledge. [71; 76; 61] explore dynamic adapter expansion/composition to allocate capacity for new tasks.

### 3 Preliminaries

#### 3.1 Cross-Domain Incremental Object Detection

Formally, given a sequence of training tasks $\{\mathcal{D}_1, \ldots, \mathcal{D}_t\}$, each phase $t$ provides a task $\mathcal{D}_1 = \{(x_n, y_n)\}$, where $x_n$ are $n$ samples drawn from domain $\mathcal{P}_t$. The corresponding labels $y_n$ belong to the label space $C_t$. During phase $t$, only the classes in $C_t$ are annotated, and the label spaces are disjoint across phases, i.e., $C_t \cap C_{t'} = \oslash$ for $t \neq t'$. The detector is sequentially updated in each phase to recognize the new classes in $C_t$ based on $\mathcal{D}_t$. After completing phase $t$, it is expected to detect all seen classes, i.e., $C_{1:t} = C_{1:(t-1)} \cup C_t$. Unlike conventional IOD, which assumes a fixed domain across all phases, i.e., $\mathcal{P}_t = \mathcal{P}_{t'}$ for all $t \neq t'$, CDIOD considers a more realistic setting that encompasses both intra-domain $\mathcal{P}_t = \mathcal{P}_{t'}$ and cross-domain scenarios $\mathcal{P}_t \neq \mathcal{P}_{t'}$.

#### 3.2 Understanding the Challenges of CDIOD

Based on GroundingDINO [39], a widely adopted VLM for IOD [6; 63], we reproduce several IOD methods to examine the challenges in CDIOD. Among them, GCD adopts full fine-tuning combined with KD to preserve prior knowledge. MD-DETR and Zira are PEFT-based methods that freeze backbone parameters. MD-DETR uses prompt pools, and Zira leverages reparameterization to expand capacity. The training process proceeds sequentially, PascalVOC(4 phases) $\rightarrow$ RUOD(2 phases) $\rightarrow$ DIOR(2 phases).

**Intra-domain Stability:** We measure the average forgetting within a domain as $\frac{1}{N-1}\sum_{i=1}^{N-1}(W_i^i - W_i^N)$, where $N$ is the number of learning phase in this domain, $W_i^i$ is the immediate performance on task $i$, and $W_i^N$ is the performance after $N$ learning phases. As shown in Fig. 2(a), VLMs

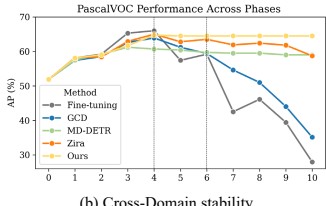

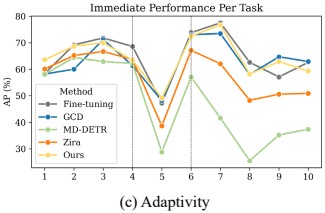

Figure 2: Stability and adaptivity analysis of different Methods on CDIOD. (a) Intra-domain stability is evaluated as forgetting percentage within each dataset, where a lower value represents less forgetting. (b) Cross-domain stability is evaluated by tracking PascalVOC performance across all training phases; vertical dashed lines denote domain transitions. (c) Adaptivity is measured by the immediate performance on each task right after its training, indicating how well the model adapts to newly introduced tasks.

show minimal forgetting on PascalVOC. However, domains with larger distribution shifts from pre-training (e.g., DIOR and RUOD) suffer severe forgetting.

**Cross-domain Stability:** We track the performance of a dataset (e.g., PascalVOC) across all learning phases to assess cross-domain stability. In Fig. 2(b), PascalVOC sub-tasks are learned sequentially (phases 1-4), achieving peak performance at phase 4. After phase 4 and phase 6, training shifts to other domains. Large domain gaps lead to substantial VLM forgetting. GCD struggles to prevent forgetting in these scenarios. Zira and MD-DETR, which maintain stable representations by freezing pre-trained parameters, better preserve prior knowledge during cross-domain cases.

**Adaptivity:** Evaluating stability alone is insufficient, as methods with high stability (e.g., zero-shot) but lack of capacity to adapt to downstream tasks typically result in suboptimal performance. We measure adaptivity by the immediate performance on each task right after training. Full fine-tuning methods generally offer stronger adaptivity. PEFT-based methods achieve comparable performance to fine-tuning on tasks near the pre-training distribution but show limited adaptivity to significant domain shifts, performing notably worse than full fine-tuning in such cases.

**Discussion:** While VLMs demonstrate strong generalization capabilities, they still suffer from significant forgetting in more generalized incremental settings. Full fine-tuning approaches offer high adaptivity but struggle to retain knowledge when faced with large domain shifts. In contrast, PEFT-based methods exhibit stronger stability by preserving pre-trained representations, yet lack sufficient adaptivity when handling out-of-distribution tasks. Balancing stability and adaptivity remains an open challenge for CDIOD.

## 4 METHOD

### 4.1 OVERVIEW

Adapter modules such as LoRA [19] and Adapters [15] improve adaptivity but remain prone to catastrophic forgetting in incremental scenarios. To enhance stability, task-specific adapters can be integrated via task-wise routing/retrieval [7; 71; 76]. This practice, however, faces performance bottlenecks due to inaccurate task-ID prediction, particularly when applied to complex CDIOD tasks (see Fig. 4). To overcome these limitations, we propose a novel framework $D^3$ to transform task-wise to robust group-wise routing. As shown in Fig. 3, we first introduce Dynamic Task Grouping (DTG) to produce a task-to-group assignments. Build on this, Dynamic Adapter Assignment (DAA) manages task-specific adapters at the group level through Incremental Group Adapters (IGA) and Intra-Group Consolidation (IGC). A dynamic training pipeline then adjusts its learning strategy based on group assignments to balance stability and adaptivity. Finally, during inference, DTG performs robust group-wise routing for each input.

### 4.2 DYNAMIC TASK GROUPING

Task distributions are commonly used for task ID inference. We extend this by leveraging distribution similarity not just for identification, but for grouping: tasks with similar distributions are

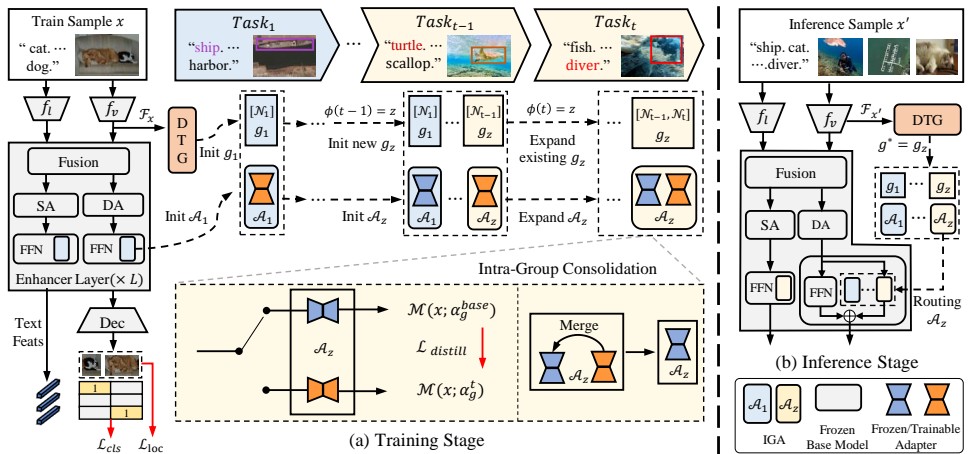

Figure 3: Overview of the D$^3$ framework. SA: Self-Attn, DA: Deform-Attn, $f_l$: language backbone, $f_v$: vision backbone. At training, given $Task_x$, the DTG estimates its task distribution $\mathcal{N}_x$ using $F_x$ and computes similarity to existing groups. The task is either assigned to the closest group $g^* = \phi(x)$ or a new group is initialized. The training pipeline branches accordingly: (1) For new groups (e.g., $Task_{t-1}$), a new IGA $\mathcal{A}_z$ is trained; (2) For existing groups (e.g., $Task_t$), $\mathcal{A}_z$ expands with a new adapter $\alpha_g^t$ initialized from $\alpha_g^{base}$. Intra-group consolidation then occurs through KD where student output $\mathcal{M}(x; \alpha_g^t)$ are aligned with teacher output $\mathcal{M}(x; \alpha_g^{base})$, followed by adapter merging. At inference, DTG performs group routing by comparing the test sample distribution $\mathcal{N}_{x'}$ to stored group ones. If $g^* = g_z$, the corresponding $\mathcal{A}_z$ is activated for prediction.

clustered to enable knowledge sharing, while dissimilar ones are isolated to prevent interference. We thus introduce DTG, an adaptive grouping mechanism based on distribution similarity. Formally, given a task sequence $\mathcal{S} = \{1, \ldots, Z\}$, we define a mapping function $\phi(\mathcal{S}) \to \{g_1, \ldots, g_z\}$ with $z \leq Z$, inducing a task partition:

$$\mathcal{S} = \bigcup_{i=1}^{z} g_i, \quad \text{where} \quad g_i \cap g_{i'} = \emptyset \quad \text{for } i \neq i'. \tag{1}$$

DTG functions as a domain discriminator which can be implemented as an autoencoder or via probabilistic modeling. In practice, we implement it as the latter due to memory efficiency. Given a new task $\mathcal{D}_t$, we extract features $\mathcal{F}_t$ using the frozen image backbone and estimate its statistics, denoted as $\mu_t = \mathbb{E}(\mathcal{F}_t)$ and $\Sigma_t = \mathbb{V}ar(\mathcal{F}_t)$. This defines a Gaussian approximation $\mathcal{N}_t = \mathcal{N}(\mu_t, \Sigma_t)$ for task $\mathcal{D}_t$. We compare $\mathcal{N}_t$ with the distribution of each task assigned to a group. For group $g$ with task set $\{k \in g\}$, the similarity is computed as the minimum KL divergence:

$$\mathrm{KL}(t, g) = \min_{k \in g} \left[ D_{KL}(\mathcal{N}_t \| \mathcal{N}_k) \right] \tag{2}$$

Let $g^* = \arg \min_g \mathrm{KL}(t, g)$ denote the group yielding the minimum KL divergence. The grouping decision follows:

$$\phi(t) = \begin{cases} g^* & \text{if } \mathrm{KL}(t, g^*) < \tau \\ \text{Init new group} & \text{otherwise} \end{cases} \tag{3}$$

where $\tau$ is the expansion threshold. If no existing group exhibits sufficient similarity to task $\mathcal{D}_t$, a new group is created to accommodate this task.

### 4.3 DYNAMIC ADAPTERS ASSIGNMENT

Building on the task-to-group allocation from DTG, we reorganize task-specific adapters into group-specific ones. This approach offers two key advantages: it effectively replaces task-wise routing with a more robust group-wise mechanism, and it enables knowledge reusing among adapters within the same group. We realize this through two key components: Incremental Group Adapters (IGA) and Intra-Group Consolidation (IGC). IGA serves as an expandable adapter module tied to each group. IGC further enhances knowledge reuse and parameter growth control within IGA.

**Incremental Group Adapters.** Each group $g_i$ is associated with an IGA, denoted as $\mathcal{A}_i$, which is initialized upon the arrival of its first task. The $\mathcal{A}_i$ comprises a set of task-specific adapters $\alpha_{g_i}^k$, one for each task $k \in g_i$. These adapters follow the LoRA [19] design and are inserted into the feed-forward networks (FFN) of both text and image branches within each enhancer layer. Denoting the FFN input as $h$, the output becomes:

$$\text{FFN}(h) + \sum_{k \in g_i} m_k \cdot B_k A_k h, \tag{4}$$

where $A_k$ and $B_k$ are LoRA parameters of task $k$, and $m_k$ is a one-hot mask indicating the active adapter.

**Intra-Group Consolidation.** We manage to model multiple tasks within single IGA module $\mathcal{A}_g$ to enable group-wise routing. But we still faces two key issues: how to effectively reuse previously learned knowledge and how to prevent the linear growth of adapter parameters. To address the aforementioned issues, we propose a per-task parameter consolidation strategy. First, to retain group-level knowledge, we employ KD from the base to the new adapter via a lightweight switching mechanism, avoiding the need to cache previous models. Each IGA $\mathcal{A}_g$ includes a base adapter $\alpha_g^{\text{base}}$. When a new task $t$ arrives (i.e., $\phi(t) = g$), a task-specific adapter $\alpha_g^t$ is initialized from the base one. We denote the model output as $\mathcal{M}(x; \alpha_g)$, where $x$ is the input sample, and $\alpha_g$ is the currently activated adapter within IGA $\mathcal{A}_g$. During training, we obtain (i) the student output using $\alpha_g^t$, and (ii) the teacher output using $\alpha_g^{\text{base}}$. The distillation loss is formally defined as:

$$\mathcal{L}_{\text{distill}} = \mathcal{L}\left(\mathcal{M}(x; \alpha_g^t), \ \mathcal{M}(x; \alpha_g^{\text{base}})\right), \tag{5}$$

where $\mathcal{L}$ refers to a topology-based KD loss. Then, to prevent parameter growth over time, we consolidate adapters within each $\mathcal{A}_g$ after training. Specifically, the new adapter $\alpha_g^t$ is merged into base adapter $\alpha_g^{\text{base}}$ via a weighted sum:

$$\alpha_g^{\text{base}} \leftarrow \lambda \alpha_g^{\text{base}} + (1 - \lambda) \alpha_g^t, \tag{6}$$

where $\lambda \in [0, 1]$ balances prior knowledge preservation and task-specific adaptation. This merging serves as parameter-level regularization which prevents parameters overly drift from the base one. We use a small $\lambda$ to favor updated representations while maintaining stability. The update is applied to both LoRA matrices $(A, B)$, after which $\alpha_g^t$ is discarded. Further details are provided in Sec. B.

### 4.4 Training and Inference

**Dynamic training pipeline.** To ensure a proper stability-adaptivity balance, $D^3$ dynamically adjusts its training pipeline based on the task assignment. It adopts a two-fold scheme: for a task assigned to a new group, the model is updated directly without a constraint term, favoring adaptivity to novel tasks. Conversely, for a task assigned to an existing group, we apply KD to retain group knowledge and mitigate label conflicts via pseudo-labeling, which prioritizes stability. By defining a binary indicator $\delta(t)$, where $\delta(t) = 1$ if task $t$ is assigned to an existing group, and $\delta(t) = 0$ otherwise, we unify the training objective as follows:

$$\mathcal{L} = \mathcal{L}_{\text{cls}} + \mathcal{L}_{\text{loc}} + \delta(t)\mathcal{L}_{\text{distill}}, \tag{7}$$

where $\mathcal{L}_{\text{cls}}$ utilizes focal loss [38], and $\mathcal{L}_{\text{loc}}$ employs L1 and GIoU losses [54].

**Group Routing for Inference.** At inference time, for a given test image $x$, we first extract its feature statistics using the backbone, defining a Gaussian approximation $\mathcal{N}_x$. The DTG identifies the most similar group $g^*$ via minimum KL divergence. If $\text{KL}(x, g^*)$ is below an out-of-distribution threshold, the corresponding $\mathcal{A}_{g^*}$ is activated across all relevant layers for prediction. Otherwise, the model defaults to zero-shot inference using the base model alone.

## 5 Experiments

**Datasets and Metrics.** We construct CDIOD using three diverse datasets covering 50 classes: DIOR [31] (remote sensing, 20 classes), Pascal VOC 2012 [9] (natural scenes, 20 classes), and RUOD [11] (underwater, 10 classes). To assess generality, we also evaluate on two additional

Table 1: CDIOD results ($AP\%$) under 0–5 and 0–10 settings. We report performance over three datasets after last training phase. All methods are based on Grounding-DINO-T. The Best and second results are shown in **bold** and underline, respectively.

| Method | 0-10 (5 phases) | | | | 0-5 (10 phases) | | | |
| --- | --- | --- | --- | --- | --- | --- | --- | --- |
| | DIOR | PascalVOC | RUOD | Average | DIOR | PascalVOC | RUOD | Average |
| Joint | 69.5 | 72.0 | 64.3 | 68.6 | 69.5 | 72.0 | 64.3 | 68.6 |
| Fine-tuning | 32.8 | 44.4 | 31.1 | 36.1 | 19.0 | 34.9 | 19.9 | 24.6 |
| CL-DETR [41] | 55.6 | 46.3 | 40.9 | 47.6 | 51.6 | 30.5 | 33.1 | 38.4 |
| MD-DETR [2] | 35.2 | 59.6 | 48.5 | 47.8 | 29.5 | 58.5 | 36.8 | 41.6 |
| GCD [63] | 56.6 | 51.7 | 44.8 | 51.0 | 52.4 | 42.0 | 36.6 | 43.7 |
| Zira [6] | 36.8 | 62.6 | 45.3 | 48.2 | 27.5 | 61.3 | 39.7 | 42.8 |
| Ours | **63.2** | **68.4** | **62.5** | **64.7** | **58.8** | **65.3** | **56.7** | **60.2** |

Table 2: IOD results ($AP\%$) on COCO 2017. Performance for base classes $C_1$, new classes $C_{2:t}$, and all classes $C_{1:t}$ are reported, denoted as 'old', 'new', and 'all', respectively.

| Method | 40-40 (2 phases) | | | 40-10 (5 phases) | | |
| --- | --- | --- | --- | --- | --- | --- |
| | old | new | all | old | new | all |
| Joint | 61.8 | 54.1 | 57.9 | 61.8 | 54.1 | 57.9 |
| Fine-tuning | 56.8 | **53.6** | 55.2 | 56.2 | 46.2 | 51.2 |
| CL-DETR | 57.6 | 52.6 | 55.1 | 51.6 | 47.6 | 49.6 |
| GCD | 58.7 | 52.4 | **55.5** | 55.4 | 48.0 | 51.7 |
| MD-DETR | 52.6 | 50.1 | 51.3 | 51.3 | 40.1 | 45.7 |
| Zira | 58.0 | 49.6 | 53.8 | **57.0** | 46.8 | 51.9 |
| Ours | **59.8** | 50.8 | 55.3 | 56.8 | **48.4** | **52.6** |

benchmarks: conventional IOD based on COCO [37] and IVLOD [6] based on ODinW-13 [30]. We use standard COCO metrics: $AP$ and $AP_{50}$.

**Experiment Setup.** CDIOD is constructed by sequentially combining the above datasets in a class-incremental manner. Each dataset is split into subsets following standard class splits [10; 24; 25], ensuring disjoint label spaces across phases. Since images may include objects from unseen classes, they can reappear across phases, reflecting realistic conditions. The setup is denoted as $N_{base} - N_{inc}$, where $N_{base}$ denotes the number of classes introduced in the initial step, and $N_{inc}$ specifies the number introduced in each subsequent phase. When $N_{base} = 0$, classes are evenly divided in each phase. We report joint training results as an upper bound (denoted Joint). We adopt two settings: 0-10 (5 phases) and 0-5 (10 phases). For example, the 0-10 setup follows DIOR (2 phases) $\rightarrow$ PascalVOC (2) $\rightarrow$ RUOD (1). After completing all phases, models are evaluated jointly across all three datasets. We report the average performance after three runs with shuffled dataset orders. Detailed description of the benchmark can be found in Sec. A.2.

**Implementation Details.** All models are built on Grounding-DINO-T, pre-trained on Objects365 [56], GoldG [32], and Cap4M [32]. Training is conducted on 8 RTX 3090 GPUs with total batch size 16. Only LoRA parameters are updated, we set the rank as 16. The learning rate is set to $1e-3$ for 11 epochs and $1e-4$ for the last. The expansion threshold is 150, and the merge factor is 0.2 for both A and B matrices.

## 5.1 RELATED BENCHMARKS

**Incremental Object Detection (IOD).** We evaluate IOD on COCO under two common settings: 40-40 and 40-10, utilizing the same methods as in our CDIOD evaluations.

**Incremental Vision-Language Object Detection (IVLOD).** For the IVLOD benchmark [6], we follow its full-shot task-incremental setup, where evaluation is confined to a task-specific label space. We employ group routing to infer the optimal group id for each input. Models are sequentially trained on ODinW-13 and evaluated after all phases. Baseline results for TFA [62], iDETR [8], AT [18], OW-DETR [14], CL-DETR [41], and Zira [6] are taken from [6].

Table 3: IVLOD full-shot results ($AP\%$) on ODinW-13 and ZCOCO (zero-shot results on COCO).

| Methods | ZCOCO | Avg | Ae | Aq | Co | Eg | Mu | Pa | Pv | Pi | Po | Ra | Sh | Th | Ve |
|---|---|---|---|---|---|---|---|---|---|---|---|---|---|---|---|
| Zero-shot | 47.4 | 46.7 | 19.1 | 20.8 | 64.7 | 57.0 | 25.4 | 54.5 | 54.8 | 66.0 | 22.1 | 62.2 | 32.8 | 70.6 | 57.2 |
| TFA | 31.0 | 47.9 | 23.8 | 30.7 | 67.2 | 61.8 | 30.5 | 50.2 | 47.7 | 60.9 | 29.3 | 61.7 | 31.4 | 66.2 | 61.6 |
| iDETR | 37.3 | 58.7 | 32.6 | 46.7 | 71.0 | 68.6 | 55.3 | 58.9 | 64.5 | 71.0 | 50.3 | 63.3 | 39.2 | 77.1 | 64.8 |
| AT | 42.3 | 51.1 | 23.6 | 39.9 | 72.3 | 65.5 | 31.5 | 50.5 | 60.5 | 66.1 | 39.1 | 53.5 | 34.0 | 68.1 | 60.2 |
| OW-DETR | 31.2 | 55.6 | 28.5 | 43.8 | 70.5 | 67.8 | 43.8 | 56.8 | 63.1 | 69.5 | 45.2 | 59.0 | 37.0 | 74.4 | 63.2 |
| CL-DETR | 32.2 | 57.3 | 29.4 | 45.2 | 71.9 | 69.9 | 45.2 | 58.5 | 65.1 | 71.7 | 46.6 | 60.8 | 38.1 | 76.7 | 65.2 |
| ZiRa | 46.1 | 59.7 | 32.8 | 48.2 | 70.3 | 69.7 | 59.3 | 58.1 | 64.0 | 70.7 | 50.1 | 67.5 | 45.5 | 76.8 | 63.5 |
| Ours | 46.4 | 60.9 | 40.3 | 54.8 | 60.5 | 78.3 | 42.6 | 59.1 | 58.4 | 75.9 | 56.1 | 69.5 | 50.5 | 79.4 | 66.4 |

## 5.2 MAIN RESULTS

**Results on CDIOD.** We evaluate our method on the CDIOD benchmark under 5 phases and 10 phases settings, where each phase introduces 10 and 5 classes, respectively (Tab. 1). Existing methods generally struggle to balance adaptivity and stability. Full fine-tuning methods adapt well to new tasks but suffer from severe forgetting, especially under longer incremental sequences. In contrast, PEFT-based IOD methods maintain more stable performance on in-distribution tasks (e.g., PascalVOC) but show limited adaptivity to new domains. Our method achieves a better balance across all domains and phases. It surpasses prior SOTA by +13.7 AP (5 phases) and +16.5 AP (10 phases), maintaining stability while remaining adaptive to tasks with distribution shifts. More detailed results per run and the impact of training order are provided in Sec. C.1.

**Results on IOD.** As shown in Tab. 2, we further evaluate our method on COCO 2017 under 2 phases and 5 phases incremental settings. For conventional IOD, simple fine-tuning already performs on par with previous SOTA, indicating that pre-trained representations inherently offer strong forgetting resistance in in-domain scenarios. Our proposed method achieves superior performance across both incremental scenarios, particularly in longer learning phases (40-10), which demonstrates its enhanced intra-domain stability.

**Results on IVLOD.** As shown in Tab. 3, our method drops 1.0 AP on ZCOCO compared to the zero-shot upper bound. It also improves the ODinW13 average by 1.2 AP over the prior SOTA, achieving the top score on 10 of the 13 datasets. These results demonstrate our method's robust downstream adaptation while effectively preserving zero-shot capability.

## 5.3 ABLATION STUDY

Table 4: Impact of different components, reporting Extra Parameters Percentage (EPP, %) and Average performance ($AP\%$) under 0-5 (10 phases) CDIOD setting.

| # | Method | EPP | DIOR | VOC | RUOD | Avg |
|---|---|---|---|---|---|---|
| 1 | Base Model | 0.00% | 2.7 | 51.9 | 19.6 | 24.7 |
| 2 | LoRA | 0.40% | 3.2 | 42.5 | 42.9 | 29.5 |
| 3 | T-LoRA | 4.00% | 41.0 | 63.7 | 49.7 | 51.5 |
| 4 | 3 + Merge | 4.00% | 29.7 | 61.5 | 40.1 | 43.8 |
| 5 | G-LoRA | 1.20% | 48.7 | 65.2 | 49.1 | 54.3 |
| 6 | 5 + Group Init | 1.20% | 51.6 | 66.7 | 51.9 | 56.7 |
| 7 | 6 + Dynamic | 1.20% | 58.8 | 65.3 | 56.7 | 60.2 |

Table 5: Impact of the expansion threshold $\tau$ on task grouping and performance under 0-5 (10 phases) setting.

| Threshold | Groups | DIOR | VOC | RUOD | Avg |
|---|---|---|---|---|---|
| $\tau = 1$ | 10 | 41.0 | 63.7 | 49.7 | 51.5 |
| $\tau = 50$ | 6 | 58.8 | 64.8 | 53.2 | 58.9 |
| $\tau = 100$ | 4 | 59.7 | 66.1 | 53.1 | 59.6 |
| $\tau = 150$ | 3 | 58.8 | 65.3 | 56.7 | 60.2 |
| $\tau = 250$ | 3 | 58.6 | 65.3 | 57.7 | 60.5 |
| $\tau = 400$ | 3 | 57.9 | 65.2 | 57.0 | 60.0 |
| $\tau = 500$ | 3 | 58.4 | 65.1 | 57.2 | 60.2 |
| $\tau = 600$ | 3 | 57.7 | 65.3 | 57.6 | 60.2 |
| $\tau = 1000$ | 2 | 23.5 | 65.2 | 57.3 | 48.7 |

**Impact of each component.** We conduct ablations under 10 phases setting to assess each component's impact (Tab. 4). Row 1 shows that pre-trained VLMs poorly generalize to remote sensing and underwater domains. A single LoRA (Row 2) shows adaptivity but suffers from severe forgetting. T-LoRA (Row 3) trains task-specific LoRA and combines them via task-wise routing, causing linear parameter growth and routing errors. Row 4 further merge all LoRA weights per task through average weighted sum, the knowledge gaps between domains leads to significant performance decline. Row 5 we introduce DTG and Eq. (6) to train Group-wise LoRA (G-LoRA) which builds a strong baseline. Knowledge from different domains are managed by groups, thus we could merge relevant LoRA weights to avoid linear parameter growth. Row 6 we replace random init with group

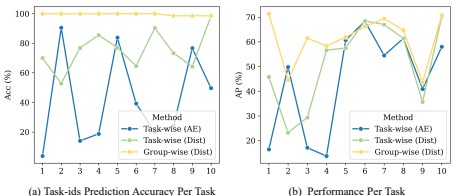
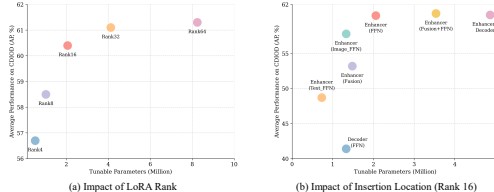

(a) Task-ids Prediction Accuracy Per Task     (b) Performance Per Task     (a) Impact of LoRA Rank     (b) Impact of Insertion Location (Rank 16)

Figure 4: Task ID prediction accuracy analysis, evaluated after 10 phases of training.

Figure 5: Analysis of LoRA Rank choice and corresponding insertion position with rank 16.

initialization, which effectively alleviate catastrophic forgetting. Our full method (Row 7) integrates a dynamic training pipeline to automatically balance adaptivity and stability, achieving balanced performance across all tasks.

**Impact of Threshold $\tau$ on Task Grouping.** We examine how the expansion threshold $\tau$ affects group numbers and performance in the 10 phases setting. A small threshold (e.g., $\tau = 1$) degenerates to task-wise routing, creating a separate IGA for each task and failing to exploit task similarities. In contrast, a large one (e.g., $\tau = 1000$) merges diverse tasks into a single group, leading to severe task interference and degraded cross-domain stability. Moderate values of $\tau$ (e.g., 100–600) allow semantically similar tasks to be grouped together, yielding the best overall performance. Notably, the framework's performance remains stable across this broader threshold range, which indicates a low sensitivity to this hyperparameter under CDIOD. To further validate this, we adopt a fixed expansion threshold ($\tau = 150$) for all comparison experiments without extra tuning. The consistent performance gains confirm practicality.

**Task Routing Accuracy Analysis.** We investigate the effect of different task ID inference strategies on routing accuracy under 10 phases setting. Upon completing all phases, we assess the routing accuracy for each task. As illustrated in Fig. 4(a), task-wise routing based on auto-encoders (Blue) suffers from severe confusion among tasks within the same domain. Distribution-based (Green) inference alleviates intra-domain confusion but still achieves suboptimal accuracy. Furthermore, Fig. 4(b) reveals that the performance of each task is highly dependent on its inference accuracy. In contrast, DTG allows for group-wise routing (Yellow), significantly reduces routing errors and consistently outperforms task-wise methods across all tasks, validating its robustness and effectiveness.

**Expert Rank and Insert Position.** We analyzed the performance and parameter efficiency of Incremental Group Adapters (IGA) by varying their insertion position and LoRA rank. "Fusion" refers to the enhancer's fusion layers, while "FFN" denotes the feed-forward networks in the enhancer's text and image branches. As shown in Fig. 5(a), inserting IGA into the FFNs (both image and text branches) provides the best balance of parameters and performance. Furthermore, Fig. 5(b) reveals that a LoRA rank of 16 achieves the most favorable trade-off between parameters and performance.

## 6 CONCLUSION AND LIMITATION.

In this work, we demonstrate that existing benchmarks inadequately capture the incremental learning challenges faced by pre-trained VLMs in real-world scenarios. To bridge this gap, we introduce CDIOD, a more generalized, domain-diverse benchmark. Experiments on CDIOD reveal that VLMs suffer from severe forgetting, and current methods fail to effectively balance adaptivity and stability in these cross-domain incremental scenarios. To address this, we propose $D^3$, which integrates IGA guided by DTG to separate tasks with distinct distributions while clustering similar ones, preventing interference and enhancing knowledge sharing. Within each group, a consolidation mechanism merges task-specific adapters, effectively controlling parameter growth. And a dynamic training pipeline is introduced to better balance stability and adaptivity. During inference, group-wise routing ensures activating the optimal IGA for each input. Extensive Experiments on three benchmarks demonstrate our method's effectiveness. However, a limitation is that DTG relies on accurate task distribution estimation, which can be unreliable when data is scarce, leading to suboptimal grouping. As a preliminary exploration of CDIOD, our work highlights the broader challenge of achieving incremental learning across diverse downstream detection tasks.

## 7 ETHICS STATEMENT

The authors of this paper have read and adhere to the ICLR Code of Ethics. We believe the work presented in this paper poses no foreseeable ethical concerns.

## 8 REPRODUCIBILITY STATEMENT

To guarantee reproducibility, we have made the following efforts. (1) The pseudo-code for our method is provided in algorithm 1. (2) All datasets used in this work are publicly available, and we provide a detailed description in Sec. A.1. Furthermore, details on the proposed benchmark are provided in Sec. A.1. (3) We have also included the experiment hyperparameter settings in Sec. B.3 and the training order in Sec. C.1. (4) Additionally, all source code required for conducting and analyzing the experiments will be made publicly available upon the paper's publication.

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

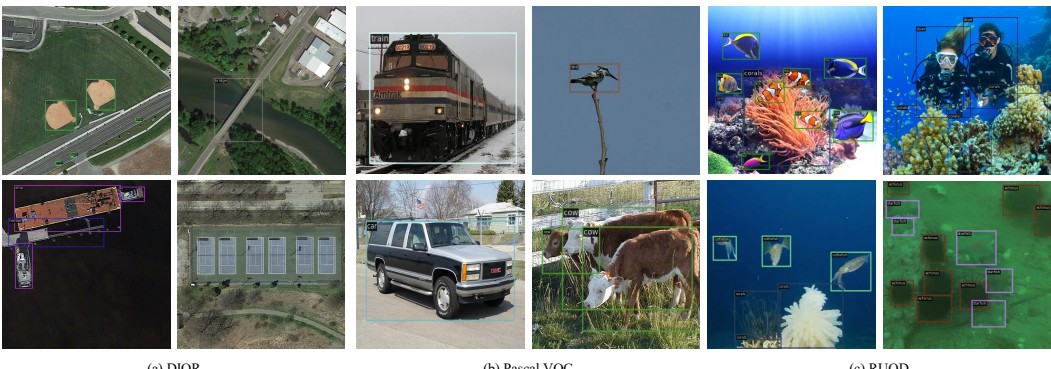

(a) DIOR            (b) Pascal VOC            (c) RUOD

Figure 6: Dataset visualization of Cross-Domain Incremental Object Detection

In this appendix, we provide: (1) Additional experiment details. (2) Additional method details. (3) More comparison and ablation results of our method. (4) Some visualization results.

## A  ADDITIONAL EXPERIMENTAL DETAILS

This section presents: (i) detailed statistics of the datasets, (ii) comprehensive description of the benchmark protocol construction, and (iii) a brief overview of the evaluated methods.

### A.1  DATASET DETAILS

To evaluate the continual learning capabilities of Vision-Language Models (VLMs) across a wider range of downstream task scenarios, we constructed the Cross-Domain Class-Incremental Object Detection (CDIOD) benchmark. This benchmark leverages datasets from three different domains, encompassing common natural scenes alongside remote sensing and underwater datasets that exhibit significant distributional gaps from typical pre-training data.

- **DIOR** [31] is a large-scale remote sensing object detection dataset. It contains 20 classes (e.g., `golffield`, `bridge`, `stadium`), with 18,463 training images and 5,000 validation images. Classes primarily feature architectural and infrastructural objects.
- **Pascal VOC 2012** [9] is a widely recognized natural scenes object detection benchmark. It comprises 20 classes (e.g., `car`, `person`, `cat`), with 13,690 training images and 3,422 validation images. Classes focus on common everyday objects.
- **RUOD** [11] is a dataset specifically curated for underwater object detection. It consists of 10 classes (e.g., `turtle`, `diver`, `starfish`), with 9,800 training images and 4,200 validation images. Classes include various marine life.

For related Incremental Object Detection (IOD) and Incremental Vision-Language Object Detection (IVLOD) benchmarks, we utilize:

- **COCO** [37] is a large-scale natural scenes dataset for object detection, segmentation. It features 80 classes (e.g., `person`, `car`, `dog`), with 118,000 training images and 5,000 validation images.
- **ODinW13** [30] is a comprehensive benchmark designed to evaluate zero-shot object detection performance. It aggregates 13 different sub-datasets, including: Aerial Maritime Drone (Ae), Aquarium (Aq), Cottontail Rabbits (Co), Egohands (Eg), Mushrooms (Mu), Packages (Pa), Pascal VOC (Pv), Pistols (Pi), Pothole (Po), Raccoon (Ra), Shellfish (Sh), Thermal Dogs and People (Th), and Vehicles (Ve).

## A.2 BENCHMARK CONSTRUCTION DETAILS

**Extending IOD for Modern VLMs.** Modern pre-trained vision-language models (VLMs), such as GLIP [32] and Grounding DINO [39], are designed to operate across diverse domains rather than being limited to a single scenario. For example, to assess their zero-shot generalization capability, these models are commonly evaluated on heterogeneous benchmarks like ODinW35, which span a wide range of domains.

We construct CDIOD not merely as a dataset extension to conventional IOD, but as a broader generalization of the IOD setting. A straightforward approach would be to retain the standard IOD setting and treat each dataset as an independent incremental process for evaluation. However, this still restricts the detection model to incremental learning within a single domain context. We further view the incremental learning process as a natural extension of pre-training, aimed at progressively expanding the model's knowledge capacity through continual learning. Therefore, we integrate datasets from different domains into a unified incremental learning protocol, enabling the model not only to perform incremental learning within a specific domain but also to acquire cross-domain continual learning ability. Ultimately, our goal is to develop a single, unified detector capable of handling diverse downstream object detection tasks across multiple domains.

**Dataset Split.** To simulate the continual learning process within a single domain, we follow the dataset splitting strategy commonly adopted in conventional IOD. Specifically, each dataset is partitioned into multiple incremental tasks based on disjoint class subsets. For example, in Pascal VOC, we split the 20 object categories into four sequential tasks, each containing five classes. Similarly, all other datasets are divided into tasks of five classes per stage. After all training phases, the model is evaluated on the full validation sets of all three datasets. This protocol thus captures both **intra-domain continual learning** (within a single dataset) and **cross-domain continual learning** (by transitioning between datasets). We assume that the model completes continual learning within one domain before being transferred to the next. While randomly shuffling all tasks across domains could present a more challenging setting, we argue that the current protocol aligns better with realistic application scenarios.

In our experiments, we report the average performance after three runs with shuffled training orders. In practice, to ensure fair comparisons, these orders were defined as **Order 1** (DIOR $\to$ Pascal VOC $\to$ RUOD), **Order 2** (Pascal VOC $\to$ RUOD $\to$ DIOR) and **Order 3** (RUOD $\to$ Pascal VOC $\to$ DIOR). Specifically, we treat the 50 classes from DIOR (20 classes), Pascal VOC (20 classes), and RUOD (10 classes) as a single, complete continual learning process. We take **Order 1** to explain the training process: Classes $1 \sim 20$ correspond to DIOR, $21 \sim 40$ to Pascal VOC, and $41 \sim 50$ to RUOD. On this basis, we consider the following two configurations:

- **0-10 (5 phases):** Each phase introduces 10 new classes, followed by evaluation on all learned classes. The training and testing process is as follows:
  - Phase 1: DIOR (classes $1 \sim 10$)
  - Phase 2: DIOR (classes $11 \sim 20$)
  - Phase 3: Pascal VOC (classes $21 \sim 30$)
  - Phase 4: Pascal VOC (classes $31 \sim 40$)
  - Phase 5: RUOD (classes $41 \sim 50$)
  - Test: DIOR + Pascal VOC + RUOD (classes $1 \sim 50$)
- **0-5 (10 phases):** Each phase introduces 5 new classes, with evaluation conducted after each phase across all accumulated classes.

### A.3 Evaluated Method Details

**Implementation Details.** We conduct a comprehensive evaluation of existing Incremental Object Detection (IOD) methods. To ensure fairness and consistency, all methods are re-implemented on the Grounding DINO-T backbone, pre-trained on Objects365, GoldG, and Cap4M. All methods are trained without a memory bank.

- **CL-DETR** [41]. A full fine-tuning-based IOD method. It adopts a from-scratch Deformable-DETR and generates pseudo-labels that do not overlap with ground truth. The pseudo-label generation strategy is architecture-agnostic and was directly reproduced on the Grounding DINO backbone. Following the original paper, we select the top 10 predictions to generate pseudo-labels, setting the overlap with ground truth to not exceed 0.7. The learning rate is set to $1e-4$. A learning rate decay of 0.1 is applied to the vision backbone, while the language backbone remains frozen.

- **GCD** [63]. A full fine-tuning-based IOD method. It adopts a from-scratch Grounding-DINO and employs correspondence distillation to transfer the teacher model's responses and topological relationships. Following the official implementation, we set the pseudo threshold to 0.4. The coefficients of the correspondence distillation loss are set as $\gamma = 1, \lambda_1 = 3, \lambda_2 = 5$. The learning rate is set to $5e-5$, with a learning rate decay of 0.1 applied to the vision backbone. The language backbone is frozen.

- **MD-DETR** [2]. A PEFT-based IOD method. It was originally initialized from a Deformable-DETR pre-trained on LVIS. It introduces vision prompts organized in a prompt pool, which are injected into the self-attention per decoder layer to facilitate task adaptation. For our re-implementation, we follow the official code, using 100 memory units ($N_m = 100$) with a length of 10 ($L_m = 10$) and a dimension of 256 ($D = 256$). We set $\lambda_Q = 0.01$ and $\delta_{bt} = 0.65$. Only the prompt pool and query function are updated during training, with the learning rate set to $1e-2$.

- **Zira** [6]. A PEFT-based IVLOD method. It is initialized from Grounding DINO pre-trained on Objects365, GoldG, and Cap4M. It integrates a re-parameterizable dual-branch module for task adaptation, inserted into both the language and vision backbone-to-enhancer connections within the neck. Following the official code, we set the coefficient for the Zil loss to $\lambda = 0.1$ and the learning rate decay for LLRB is $\eta = 0.2$. The learnable scale factor is initialized with $s = 0.1$. Only the RDB module is updated. The learning rate is initialized at $1e-3$ for the first epoch and then decays to $1e-4$.

- **MoE-Adapters** [71]. A PEFT-based Method for MTIL from incremental classification, originally implemented in CLIP. It introduces adapters organized by a mixture-of-experts for task adaptation and employs an activate-freeze strategy to alleviate catastrophic forgetting. For our re-implementation, we insert the MoE-Adapters into the FFN of the enhancer. Following the official code, we set the bottleneck for the adapter to $D = 64$ and use a top-2 gating strategy. For each task, we use 2 experts and 1 router, resulting in $N_E = 20$ experts and $N_R = 10$ routers in 10 phases setting. For routing, we use the mean of the image tokens instead of the [CLS] token. We set the learning rate $1e-3$ for adapter and router, $3e-3$ for domain predictor.

### A.4 Related Benchmarks

**Incremental Object Detection (IOD).** Incremental Object Detection typically indicates Class-Incremental Object Detection. Conventional IOD benchmarks typically partition a general-domain dataset such as COCO into disjoint tasks defined by category labels. Each phase introduces new object classes under the assumption of a consistent data distribution. IOD can be seen as a special case of CDIOD where the domain remains fixed.

**Domain Incremental Object Detection (DIOD).** DIOD focuses on the challenge of a model continuously adapting to a sequence of shifting domains. While the domain changes incrementally, the set of object classes is typically assumed to remain fixed. Our CDIOD benchmark presents a more complex problem by combining the challenges of both DIOD (domain shift) and IOD (class-incremental learning), requiring a method to handle both novel domains and novel classes simultaneously.

**Incremental Vision-Language Object Detection (IVLOD).** IVLOD [6] focuses on incrementally adapting pre-trained vision-language models (VLMs) to a sequence of tasks from the ODinW-13 benchmark while preserving zero-shot generalization. IVLOD primarily addresses task-incremental scenarios, where the model's predictions are confined to a task-specific class space and the label

space is allowed to overlap across tasks. In contrast, our work tackles a more complex and realistic class-incremental challenge. For training, CDIOD features a disjoint label space, 'person' labeled in task $t$ will not be labeled in subsequent tasks $t^{'} > t$. For inference, all learned classes are evaluated together without knowing which task set the test image belongs to.

**Open-Vocabulary Object Detection (OVD).** Open-Vocabulary Object Detection focuses on building models capable of detecting any category without being explicitly trained on them beforehand. The primary goal of OVD is zero-shot generalization, the ability to identify unseen categories in downstream tasks without the need for additional supervision. Starting from an generalizable OVD model, CDIOD focuses on the subsequent challenge of continual learning. CDIOD evaluates a model's ability to continuously learn from a sequence of supervised tasks, focusing on challenges of adaptation to new data and preservation of previously acquired knowledge.

**Domain Adaptation Object Detection (DAOD).** Domain Adaptation Object Detection typically focuses on a single-step adaptation process where a model trained on a labeled source domain is adapted to an unlabeled target domain. A key assumption in most DAOD methods is that the set of object classes across the source and target domains is identical. Unlike DAOD, our benchmark addresses a multi-phase, supervised continuous learning scenario where both the domain and the object classes change over time, and all new data is provided with labels.

**Cross-Domain Few-shot Object Detection (CDFSOD).** CDFSOD [12] focuses on adapting pre-trained detectors to downstream tasks that exhibit a significant domain gap with the pre-trained data under a few-shot setting. CDFSOD is a single-step adaptation task; it is concerned solely with the model's performance on the new downstream task and does not evaluate its ability to retain knowledge from the source domain afterward. In contrast, our benchmark evaluates a continuous learning process with multiple phases, requiring the model to simultaneously maintain performance on all previously learned tasks.

# B ADDITIONAL METHOD DETAILS

In this section, we provide supplementary details of our method, including: (i) Pseudo code illustrating the training and inference pipelines. (ii) A detailed formulation of DTG and KD loss (iii) Hyperparameter configurations adopted in our experiments, and (iv) Understanding of IGC.

## B.1 PSEUDO CODE.

We present the pseudo code of our proposed method in algorithm 1 and algorithm 2, detailing both the training and inference procedures.

## B.2 DETAILED DYNAMIC TASK GROUPING.

For our Dynamic Task Grouping (DTG), we model each task's feature distribution by extracting features $\mathcal{F}_t$ from the image backbone, utilizing either the final layer's output for faster inference speeds or a multi-level representation for higher accuracy(we default to the final layer). We approximate this distribution as a multivariate Gaussian $\mathcal{N}(\mu_t, \Sigma_t)$, where $\mu_t = \mathbb{E}(\mathcal{F}_t)$ is the sample mean and $\Sigma_t = \text{Var}(\mathcal{F}_t)$ is the full covariance matrix, capturing inter-dimensional correlations. For numerical stability, we regularize the covariance by adding a small identity matrix ($\Sigma'_t = \Sigma_t + \epsilon I$). Group similarity is measured using the Symmetrized KL Divergence for computational efficiency.

## B.3 DETAILED KNOWLEDGE DISTILLATION LOSS.

We denote the model output as $\mathcal{M}(x; \mathcal{A}_g, m)$, where $x$ is the input sample, $\mathcal{A}_g$ is the IGA, and $m$ a binary switching mask. During training, we compute: (i) the student output with $m_t = 1$ (activating $\alpha_g^t$), and (ii) the teacher output with $m_{\text{base}} = 1$ (activating $\alpha_g^{\text{base}}$). Our distillation loss is defined as follows:

$$\mathcal{L}_{\text{distill}} = \mathcal{L}\left(\mathcal{M}(x; \mathcal{A}_g, m_t), \ \mathcal{M}(x; \mathcal{A}_g, m_{\text{base}})\right), \tag{8}$$

In practice, $\mathcal{L}$ serves as a soft constraint term, which we implement using *topology distillation* as introduced by [63]. Specifically, we define the object prototype for class $c$ as:

$$p_c = \frac{1}{N_c} \sum_{i=1}^{N_c} \alpha_i q_i, \tag{9}$$

where $q_i$ is the output query feature (from the last decoder layer), $\alpha_i$ is the confidence score derived from the predicted logits, and $N_c$ is the number of instances in class $c$. We then compute the pairwise relation matrix over classes:

$$R_{ij} = \|p_i - p_j\|_2, \quad i,j \in C_{1:t-1}, \tag{10}$$

and define the topology loss as:

$$\mathcal{L}(\mathcal{M}(x; \mathcal{A}_g, m_t), \mathcal{M}(x; \mathcal{A}_g, m_{\text{base}})) = \|R^{\text{new}} - R^{\text{base}}\|_2, \tag{11}$$

where $R^{\text{base}}$ is obtained by switching to the base adapter. Similarly, to maintain cross-modal structural consistency. Finally, the total distillation loss is expressed as:

$$\mathcal{L}_{\text{distill}} = \gamma_1 \mathcal{L}_{\text{topology\_image}} + \gamma_2 \mathcal{L}_{\text{topology\_text}}, \tag{12}$$

where $\gamma_1$ and $\gamma_2$ are balancing coefficients controlling the contributions of image and text topology preservation, respectively.

### B.4 Hyperparameter setup.

Our training objective integrates five hyperparameters. The overall loss is formulated as:

$$\mathcal{L} = \mathcal{L}_{\text{cls}} + \mathcal{L}_{\text{loc}} + \delta(t)\mathcal{L}_{\text{distill}} \tag{13}$$
$$= \lambda_{\text{focal}}\mathcal{L}_{\text{focal}} + \lambda_{\text{L1}}\mathcal{L}_{\text{L1}} + \lambda_{\text{GIoU}}\mathcal{L}_{\text{GIoU}} + \delta(t)\mathcal{L}_{\text{distill}}, \tag{14}$$

where $\delta(t) = 1$ indicates we expand existing group, and that the distillation loss is applied. Following [39; 63], we set $\lambda_{\text{focal}} = 1$, $\lambda_{\text{L1}} = 5$, $\lambda_{\text{GIoU}} = 2$ and $\gamma_1 = 3$ and $\gamma_2 = 5$ for all experiments without aditional tuning.

Besides, for adapter merging, the merging factor $\lambda_{\text{merge}} \in [0,1]$ controls the trade-off between preserving prior knowledge and adapting to the new task. In our experiments, we set $\lambda_{\text{merge}} = 0.2$ which obatains the most balanced performance as shown in Tab. 9.

### B.5 Why Intra-Group Consolidation works.

**Group-Init.** For tasks within an existing group, we initialize a new adapter from the group's base one. This "warm-start" approach provides the model with a knowledgeable starting point that already contains information from old tasks, rather than beginning from a random state, which has been proven effective for alleviating catastrophic forgetting [27; 36]. This also aligns with the principle of linear mode connectivity (LMC) [49; 44], which states that a sharing initialization is crucial for keeping solutions of related tasks within a connected low error basin. By initializing from the base adapter, we start training already inside this optimal basin, making learning a more stable and efficient search for a nearby solution. This connectivity provides the theoretical foundation for why adapters in the same group can be linearly merged: since they exist in the same basin, their weighted average is also likely a high-performing solution.

**Group-KD.** The assumption of LMC generally holds for tasks like PASCAL VOC that are well-aligned with the pre-training distribution. In such cases, the pre-trained model has already situated the parameters within a favorable low-error basin, requiring the adapter to perform only minimal exploration to reach its optimal solution. However, for tasks that are dissimilar to the pre-training distribution (e.g., DIOR sub-tasks), their optimal solutions in the parameter space can be far from the base adapter's. In this case, a direct linear merge becomes suboptimal, as the simple average of two distant points is likely to fall into a high-error region. To address this, we leverage KD as an implicit constraint. During training, KD forces the new adapter's solution to remain functionally consistent with the base adapter, which actively pulls the two solutions closer and aligns them. This alignment ensures that the final merge is a robust consolidation of two compatible solutions.

Table 6: CDIOD results ($AP\%$) under 0–5 and 0–10 settings across three training orders. We report performance on all datasets after the final training phase.

| Training Orders | Method | 0-10 (5 phases) | | | | 0-5 (10 phases) | | | |
|---|---|---|---|---|---|---|---|---|---|
| | | DIOR | PascalVOC | RUOD | Avg | DIOR | PascalVOC | RUOD | Avg |
| | Joint | 69.5 | 72.0 | 64.3 | 68.6 | 69.5 | 72.0 | 64.3 | 68.6 |
| Order 1 | CL-DETR | 35.3 | 57.7 | 63.9 | 52.3 | 23.0 | 47.6 | 57.5 | 42.7 |
| | MD-DETR | 34.9 | 61.4 | 49.9 | 48.7 | 28.6 | 59.4 | 38.1 | 42.0 |
| | MoE-Adapters | 32.7 | 66.1 | 59.5 | 52.8 | 20.5 | 59.3 | 41.5 | 40.4 |
| | GCD | 43.2 | 60.0 | 62.8 | 55.3 | 37.0 | 52.0 | 58.3 | 49.1 |
| | Zira | 29.8 | 66.2 | 54.3 | 50.1 | 22.0 | 63.9 | 47.2 | 44.4 |
| | Ours | 64.7 | 68.2 | 62.6 | 65.2 | 58.5 | 65.5 | 57.2 | 60.4 |
| Order 2 | CL-DETR | 65.8 | 34.0 | 35.5 | 45.1 | 66.1 | 17.6 | 21.6 | 35.1 |
| | MD-DETR | 35.4 | 58.5 | 48.4 | 47.4 | 30.1 | 57.9 | 36.5 | 41.5 |
| | MoE-Adapters | 33.7 | 63.9 | 55.7 | 51.1 | 20.8 | 59.2 | 36.2 | 38.7 |
| | GCD | 63.5 | 41.5 | 40.8 | 48.6 | 61.9 | 35.1 | 27.5 | 41.5 |
| | Zira | 40.4 | 59.3 | 41.1 | 46.9 | 30.8 | 59.3 | 36.1 | 42.1 |
| | Ours | 63.6 | 68.7 | 62.3 | 64.9 | 58.9 | 64.5 | 56.7 | 60.0 |
| Order 3 | CL-DETR | 65.6 | 47.2 | 23.2 | 45.3 | 63.8 | 26.4 | 20.3 | 36.8 |
| | MD-DETR | 35.2 | 59.0 | 47.1 | 47.1 | 29.7 | 58.2 | 35.8 | 41.2 |
| | MoE-Adapters | 34.2 | 64.0 | 57.7 | 52.0 | 21.7 | 61.0 | 43.4 | 42.0 |
| | GCD | 63.1 | 53.5 | 30.9 | 49.2 | 58.2 | 39.0 | 23.9 | 40.4 |
| | Zira | 40.2 | 62.3 | 40.6 | 47.7 | 29.7 | 60.6 | 35.8 | 42.0 |
| | Ours | 61.2 | 68.4 | 62.5 | 64.0 | 58.9 | 65.8 | 56.2 | 60.3 |

Table 7: Performance and Computation Costs under 0–5 (10 phases) settings. Train and Test Params refer to parameters updated during training and activated at inference, respectively. Percentages denote ratio to base model parameters; ↑ indicates increased percentage. All methods are PEFT-based except GCD and CL-DETR.

| Method | Technical | Activation strategy | Avg | Train Params | Test Params | FLOPS |
|---|---|---|---|---|---|---|
| CL-DETR | Pseudo-label | Base model only | 38.4 | 64.2M(37.1%) | 173.1M(↑0.0%) | 464G |
| GCD | Knowledge Distillation | Base model only | 43.7 | 64.2M(37.1%) | 173.1M(↑0.0%) | 464G |
| MD-DETR | Prompt pool | Retrieval function | 41.6 | 0.28M(0.16%) | 173.2M(↑0.02%) | 465G |
| Zira | Rep Dual-branch | Fixed branch | 42.8 | 4.38M(2.5%) | 177.5M(↑ 2.5%) | 467G |
| MoE-Adapters | Mixture of Task-wise Adapters | Token-wise routing | 40.4 | 9.69M(5.6%) | 175.5M(↑1.47%) | 506G |
| T-LoRA | Task-wise LoRA | Task-wise routing | 51.5 | 6.90M(4.0%) | 173.8M(↑ 0.4%) | 473G |
| Ours | Dynamic Group-wise LoRA | Group-wise routing | 60.2 | 2.06M(1.2%) | 173.8M(↑0.4%) | 473G |

# C  ADDITIONAL RESULTS

This section presents additional comparison results including : (i) Detailed CDIOD results of different training order and computation cost comparisons (ii) per-task performance analysis, and (iii) extended ablation studies on various design choices and hyperparameters.

## C.1  ADDITIONAL COMPARISON RESULTS

**Detailed CDIOD results and Impact of training Order.** To further assess the impact of training order of tasks. we provide a detailed per order performance. As shown in Tab. 6, existing incremental methods are highly sensitive to the sequence of tasks and exhibit significant performance fluctuations. This instability is a key limitation in real-world applications where the arrival of new data is unpredictable. In contrast, our method consistently delivers stable performance across all three orders. This robustness to variations in the training sequence is a critical property for a practical incremental learning algorithm, demonstrating our framework's reliability in non-stationary and unpredictable environments.

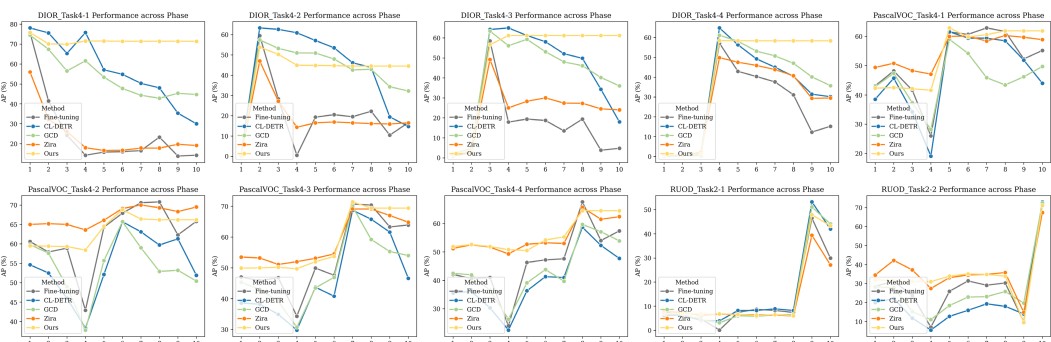

Figure 7: Performance across phases under the 0–5 (10-phase) setting with training order 1, where the x-axis denotes the training phase.

**Detailed comparison and discussion.** As shown in Tab. 7, we report the Average Performance (3 runs) and computational costs of each method under the 10-phase setting. And we provide a detailed discussion of each method.

- **CL-DETR.** It directly fine-tunes the base model using pseudo-labels, which means it has a higher number of trainable parameters but introduces no additional overhead during inference. However, under cross-domain scenarios, old classes may be absent or suffer from modality gaps. This makes it impractical to generate robust pseudo-labels.
- **GCD.** It uses KD to transfer knowledge. This approach also results in a high number of trainable parameters but introduces no overhead at inference. While the KD loss is designed to force the student model to align with the teacher's output, this alignment can be problematic in cross-domain tasks where the teacher's responses are often noisy, which may lead to catastrophic forgetting.
- **Zira.** It introduces a fixed dual-branch module attached to the base model. After learning each task, a high-learning-rate branch is merged into a low-learning-rate branch. While this design allows Zira to adapt to new tasks with a fixed parameter budget, it struggles to balance stability and adaptivity.
- **MD-DETR.** It utilizes expandable prompt pools to incrementally learn new tasks and employs a retrieval function to compose prompts with weighted sum for inference. Although this approach is memory-efficient, the expressive power of prompts limits the model's overall adaptive capability. Furthermore, its retrieval accuracy diminishes as the size of prompt pools increases.
- **MoE-Adapter.** It employs a Mixture-of-Experts (MoE) structure to combine adapters. It first uses a domain discriminator to select a task-specific router, which then performs token-wise routing to activate the corresponding adapter. When applied to object detection tasks, this design faces two critical issues. First, relying on a domain discriminator for router activation creates a bottleneck in task-ID identification. Second, the complex detection scenes make its token-wise routing susceptible to misallocation. Consequently, MoE-Adapter exhibits subpar performance despite activating a larger set of parameters.
- **T-LoRA.** It trains task-specific LoRA modules and uses task-level routing to select the optimal ones for inference. This straightforward approach achieves good performance. However, it models each task in isolation, ignoring shared knowledge and leading to a linear increase in parameters. Its primary performance bottleneck remains the reliance on accurate task-ID inference.

To overcome these limitations, our method dynamically groups multiple tasks, which enables knowledge reuse and effectively controls parameter growth. This strategy eliminates the need for precise task-ID inference, as identifying a broader task group is significantly more robust. Furthermore, this dynamic pipeline allows for the creation of new groups when facing novel tasks, enhancing adaptability, while similar tasks can be effectively integrated into existing groups to maintain stability. As shown in Table 1, our approach introduces only 1.2% additional trainable parameters and activates just 0.4% extra parameters at inference. The FLOPs increase by a mere 9G compared to the base model.

**Per-task performance across phases.** In Fig. 7, we present the detailed performance of each sub-task under the 0-5 (10 phases) setting with training order1 (DIOR → Pascal VOC → RUOD). Given the zero-shot capability of pre-trained VLMs, we evaluate all sub-tasks after each training phase to

Table 8: Impact of different components, reporting Extra Parameters Percentage (EPP, %) and Average performance ($AP\%$) under 0-5 (10 phases) CDIOD setting. Row 0 indicates Zero-shot. LoRA is set rank 16 and inserted into enhancer's FFN for all ablation results in this table.

| Row | LoRA | Task-wise | Raw Merge | DTG | Intra-Group Consolidation | | | EPP | DIOR | VOC | RUOD | Avg |
| --- | --- | --- | --- | --- | --- | --- | --- | --- | --- | --- | --- | --- |
| | | | | | Group_Merge | Group_Init | Group_KD | | | | | |
| 0 | | | | | | | | 0.00% | 2.7 | 51.9 | 19.6 | 24.7 |
| 1 | ✓ | | | | | | | 0.40% | 3.2 | 42.5 | 42.9 | 29.5 |
| 2 | ✓ | | | | | | ✓ | 0.40% | 5.1 | 52.0 | 53.8 | 37.0 |
| 3 | ✓ | ✓ | | | | | | 4.00% | 41.0 | 63.7 | 49.7 | 51.5 |
| 4 | ✓ | ✓ | ✓ | | | | | 4.00% | 29.7 | 61.5 | 40.1 | 43.8 |
| 5 | ✓ | | | ✓ | ✓ | | | 1.20% | 48.7 | 65.2 | 49.1 | 54.3 |
| 6 | ✓ | | | ✓ | ✓ | ✓ | | 1.20% | 51.6 | 66.7 | 51.9 | 56.7 |
| 7 | ✓ | | | ✓ | | | ✓ | 1.20% | 50.5 | 60.9 | 52.3 | 54.6 |
| 8 | ✓ | | | ✓ | | ✓ | ✓ | 1.20% | 55.7 | 61.3 | 55.4 | 57.5 |
| 9 | ✓ | | | ✓ | ✓ | ✓ | ✓ | **1.20%** | **58.8** | 65.3 | **56.7** | **60.2** |

Table 9: Impact of the merging factor $\lambda_{merge}$ under 0-5 (10 phases) setting.

| Threshold | DIOR | VOC | RUOD | Avg |
| --- | --- | --- | --- | --- |
| $\lambda_{merge} = 0.0$ | 55.7 | 61.3 | 55.4 | 57.5 |
| $\lambda_{merge} = 0.1$ | 58.4 | 65.2 | 56.8 | 60.1 |
| $\lambda_{merge} = 0.2$ | 58.8 | 65.3 | 56.7 | 60.2 |
| $\lambda_{merge} = 0.3$ | 55.0 | 64.5 | 57.0 | 58.8 |
| $\lambda_{merge} = 0.4$ | 47.8 | 65.9 | 55.0 | 56.2 |
| $\lambda_{merge} = 0.5$ | 39.3 | 64.8 | 49.4 | 51.2 |
| $\lambda_{merge} = 0.6$ | 31.7 | 64.1 | 44.5 | 46.8 |
| $\lambda_{merge} = 0.7$ | 23.6 | 63.3 | 35.6 | 40.8 |

Table 10: Impact of the ood threshold under 0-5 setting.

| Threshold | DIOR | VOC | RUOD | Avg |
| --- | --- | --- | --- | --- |
| $th_{ood} = 200$ | 54.2 | 64.2 | 53.4 | 57.3 |
| $th_{ood} = 300$ | 57.9 | 65.3 | 55.8 | 59.7 |
| $th_{ood} = 400$ | 58.3 | 65.5 | 56.4 | 60.1 |
| $th_{ood} = 500$ | 58.5 | 65.5 | 56.7 | 60.2 |
| $th_{ood} = 600$ | 58.8 | 65.3 | 56.7 | 60.2 |

track their performance evolution across 10 phases. This serves purely as an analytical experiment, since future tasks are unknown in practical CDIOD training. For example, with 50 total categories, DIOR_Task4-1 introduces the first 5 classes in phase 1, achieves peak performance in that phase, and gradually degrades over subsequent phases.

From the results, our method demonstrates strong adaptivity, achieving immediate performance comparable to full fine-tuning for new tasks, while maintaining stability by effectively preserving prior task performance in cross-domain incremental scenarios. Furthermore, our approach also well sustains the model's generalization ability. For instance, when PascalVOC_Task4-3 is introduced in phase 7, its zero-shot performance from phases 1 to 6 remains largely intact. This is further validated by the ZCOCO results in IVLOD benchmark.

## C.2 ADDITIONAL ABLATION RESULTS

**Detailed ablation of each components.** We provide a detailed ablation results of each component of our framework. Row 0 indicates Zero-shot performance. Row 1 represent sequentially fine-tuning LoRA, which leads to severe catastrophic forgetting. Row 2 represent fine-tune LoRA with KD loss Eq. (5), the result show that this practice still unable to overcome forgetting under cross-domain scenarios. Row 3 represent train task-specific expert module per task and combine them through task-wise routing mechanism, a design widely used in recent works [2; 71; 76]. In this case, the LoRA parameters increase linearly with task numbers, which is still suboptimal due to performance bottleneck of task-id predictions. Row 4 represent we merge LoRA weights trained on each task through averaging merging, however knowledge of different domains varies significantly which leads to poor performance.

Dynamic Task Grouping (DTG) alone just construct task-to-group mapping which can't be ablated. In Row 5, instead, we leverage DTG and Group_Merge to train Group-wise LoRA. Knowledge from different domains are managed by groups, where we could merge relevant LoRA weights. This design transforms the task-wise routing to robust group-wise routing, which builds up a strong baseline

Table 11: Comparison of adapter variants within our framework, reporting Extra Parameters Percentage (EPP, %) and Average performance ($AP\%$) under 0-5 (10 phases) CDIOD setting. Both are inserted into the enhancer's FFN.

| Method | EPP | DIOR | VOC | RUOD | Avg |
|---|---|---|---|---|---|
| LoRA(r=4) | 0.30% | 53.2 | 64.1 | 52.9 | 56.7 |
| LoRA(r=8) | 0.60% | 55.6 | 64.7 | 55.2 | 58.5 |
| LoRA(r=16) | 1.20% | 58.8 | 65.3 | 56.7 | 60.2 |
| LoRA(r=32) | 2.40% | 59.8 | 65.9 | 57.6 | 61.1 |
| Adapter(d=16) | 0.18% | 48.7 | 63.8 | 49.7 | 54.1 |
| Adapter(d=32) | 0.35% | 51.5 | 64.1 | 52.3 | 56.0 |
| Adapter(d=64) | 0.69% | 53.9 | 64.5 | 53.5 | 57.3 |
| Adapter(d=128) | 1.38% | 55.8 | 65.0 | 56.5 | 59.1 |
| Adapter(d=256) | 2.75% | 57.7 | 65.3 | 57.0 | 60.0 |

for our framework. To enhance model compositionality, we introduce group initialization mechanism. Once we expand existing group, we initialize the new task-specific adapter from the base one. Row 7 and Row 8, we also ablate the merging process to focus on the impact of Group_KD. The results show that KD is crucial for alleviating forgetting, particularly for remote sensing and underwater tasks, which are not well-aligned with the pre-trained model. Our full method (Row 9) combines all components and adopt dynamic training pipeline which automatically switch between direct adapter updates and updates with Intra-Group Consolidation (IGC), effectively balance adaptivity and stability.

**Impact of Merge Factor $\lambda_{\mathbf{merge}}$.** We investigate the effect of the merge factor $\lambda_{\mathrm{merge}}$ on model performance. A smaller value encourages the model to absorb more task-specific knowledge, whereas a larger value emphasizes the preservation of base knowledge. The merging step essentially serves as a regularization mechanism on the parameter space. As shown in Tab. 9, omitting the merging step entirely leads the model to overfit to the new task, while an excessively large $\lambda_{\mathrm{merge}}$ overly constrains model updates, both of which result in suboptimal performance. A relatively small $\lambda_{\mathrm{merge}}$ achieves better results. Therefore, we set $\lambda_{\mathrm{merge}} = 0.2$ in our experiments to strike an effective balance between knowledge retention and task-specific adaptation.

**Out-of-distribution threshold.** At inference time, **as an optional choice**, we introduce an out-of-distribution (OOD) threshold $th_{ood}$ to handle inputs that are far from any known group distribution. If an input exceeds this threshold, no IGA is activated; instead, it is processed solely by the base model, leveraging the zero-shot capability of the pre-trained VLM for unseen samples. As shown in Tab. 10, setting the OOD threshold too low risks misrouting samples from known tasks to zero-shot prediction. In contrast, using a relatively higher threshold (e.g., 600) effectively avoids this issue.

**Comparison of adapter variants.** We evaluated different adapter modules for the base adapter in our framework, comparing LoRA [19] and Adapter [4]. In our implementation, LoRA with rank r is attached to the two linear layers of the feed-forward network (FFN), while the Adapter module with bottleneck dimension d is placed outside the FFN. As shown in Tab. 11, when the number of effective parameters is comparable (e.g., LoRA(r=16) and Adapter(d=128)), LoRA consistently achieves superior performance.

# D  ADDITIONAL VISUALIZATION

## D.1  CLASS-LEVEL T-SNE VISUALIZATION.

To qualitatively assess the learned representations, we perform a T-SNE visualization on the output query features. Each data point in the plot is colored and labeled according to its predicted class. For this analysis, we sample 10 classes from each of the three datasets to generate the T-SNE plots. The top row of Fig. 8 displays the features extracted from the base model's zero-shot outputs, while the bottom row corresponds to the features learned by our method after the complete incremental process. As the figure illustrates, our method consistently produces features that are well-separated by class across all three datasets, highlighting its effectiveness in learning distinct and robust representations.

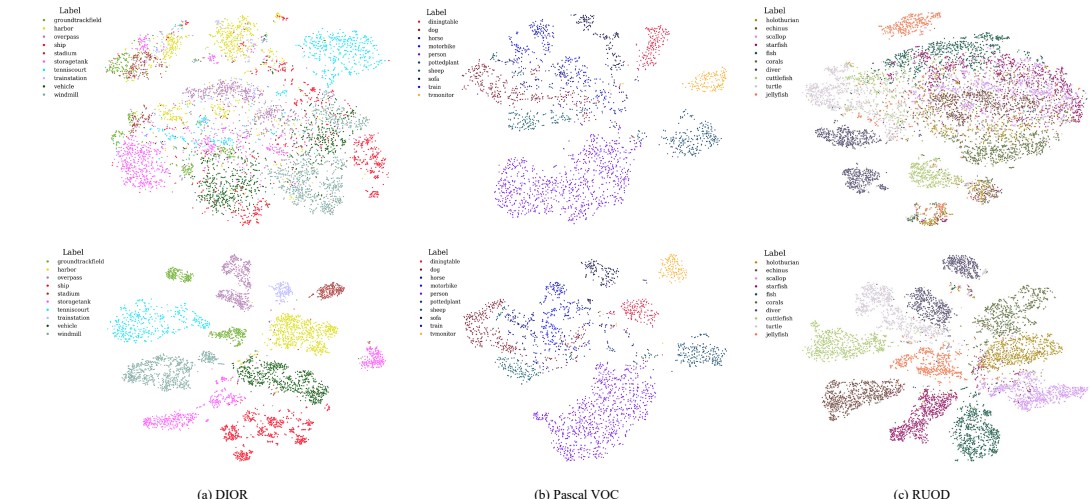

(a) DIOR        (b) Pascal VOC        (c) RUOD

Figure 8: Class-level T-SNE of output query features, with each point labeled by its predicted class. Row 1: Zero-shot; Row 2: Ours.

## D.2 QUALITATIVE RESULTS.

We provide a qualitative comparison of Zero-shot, GCD, Zira, and our proposed method under the 0-5 (10-phase) setting, with all models evaluated after completing all training phases.

- **DIOR (Remote Sensing):** The pre-trained Grounding-DINO model struggles to generalize effectively to the DIOR domain, as demonstrated by its zero-shot performance. This is likely because objects of interest in remote sensing often appear as background in its pre-training datasets (e.g., Objects365). In the incremental setting, both GCD and Zira exhibit severe catastrophic forgetting on this domain, whereas our method produces predictions that closely match the ground truth, demonstrating superior retention of DIOR-specific knowledge.
- **Pascal VOC (Natural Scenes):** For the Pascal VOC domain, which is closer to the pre-training data distribution, most methods perform well. However, GCD is a notable exception, as it suffers from significant forgetting in this cross-domain incremental scenario, highlighting its limited ability to preserve knowledge across substantial domain shifts.
- **RUOD (Underwater):** The RUOD domain presents a unique challenge due to its distinct visual characteristics. Here, Zira struggles to adapt effectively, which can be attributed to its limited adaptivity on out-of-distribution tasks.

In summary, across all evaluated tasks and domains, our method consistently demonstrates a balanced and robust performance, mitigating the forgetting issues observed in other methods and effectively adapting to diverse data distributions.

## E THE USE OF LARGE LANGUAGE MODEL

During the writing process, we utilized a large language model (LLM) to assist with editing and refining the manuscript. The LLM's role was confined to improving the fluency and grammatical correctness of the text, ensuring our arguments were presented clearly and concisely. It did not contribute to the ideation, data analysis, or core scientific content of the paper.

---

**Algorithm 1** Training Procedure

---

1: **Input:** Sequence of tasks $\{\mathcal{D}_1, \ldots, \mathcal{D}_T\}$; Model $\mathcal{M}$; $\phi$: Task to group mapping; $G$: set of groups; $\mathcal{A}_g$: Incremental Group Adapters (IGA) of group $g$.
2: **Output:** A set of trained IGA $\{\mathcal{A}_g\}_{g \in G}$.
3:
4: **for** task $\mathcal{D}_t \in \{\mathcal{D}_1, \ldots, \mathcal{D}_T\}$ **do**
5:    $\{$1. Dynamic Task Grouping$\}$
6:    Iterate $\mathcal{D}_t$ to estimate task distribution $\mathcal{N}_t$.
7:    $g^* \leftarrow \arg\min_{g \in G} \mathrm{KL}(\mathcal{N}_t \| \mathcal{N}_g)$.                             ▷ Find the most similar group $g^*$
8:
9:    $\{$2. Dynamic Adapter Assignment$\}$
10:    **if** $G$ is empty or $\mathrm{KL}(\mathcal{N}_t \| \mathcal{N}_{g^*}) \geq \tau$ **then**
11:       Initialize base adapter $\alpha_{g_{\text{new}}}^{\text{base}}$           ▷ Create a new group $g_{\text{new}}$ and its IGA $\mathcal{A}_{g_{\text{new}}}$
12:       $\alpha_{\text{active}} \leftarrow \alpha_{g_{\text{new}}}^{\text{base}}$
13:    **else**
14:       Initialize new adapter $\alpha_{g^*}^t$ from $\alpha_{g^*}^{\text{base}}$    ▷ Assign task to $g^*$ and expand its IGA $\mathcal{A}_{g^*}$
15:       $\alpha_{\text{active}} \leftarrow \alpha_{g^*}^t$
16:    **end if**
17:
18:    $\{$3. Dynamic Training Pipeline$\}$
19:    **for** each training epoch **do**
20:       **for** each batch $x \in \mathcal{D}_t$ **do**
21:          **if** task $t$ assigned to a new group **then**
22:             $\mathcal{L} \leftarrow \mathcal{L}_{\text{cls}} + \mathcal{L}_{\text{loc}}$.          ▷ Train without constraint to enhance adaptivity.
23:          **else**
24:             $\mathcal{L} \leftarrow \mathcal{L}_{\text{cls}} + \mathcal{L}_{\text{loc}} + \mathcal{L}_{\text{distill}}$.    ▷ Train with Eq. (12) to retain knowledge for stability.
25:          **end if**
26:          Update parameters of $\alpha_{\text{active}}$ using loss $\mathcal{L}$.
27:       **end for**
28:    **end for**
29:
30:    $\{$4. Merge Adapters after training$\}$
31:    **if** task $t$ was assigned to an existing group $g^*$ **then**
32:       $\alpha_{g^*}^{\text{base}} \leftarrow \lambda \alpha_{g^*}^{\text{base}} + (1 - \lambda)\alpha_{\text{active}}$.         ▷ Merge into group's base adapter
33:       Discard $\alpha_{\text{active}}$.
34:    **end if**
35: **end for**

---

**Algorithm 2** Inference Procedure

---

1: **Input:** Test image $x$; trained model $\mathcal{M}$ with frozen backbone $f_v$; group set $G$ with distributions $\{\mathcal{N}_g\}_{g \in G}$ and adapters $\{\mathcal{A}_g\}_{g \in G}$; out-of-distribution (OOD) threshold $\tau_{ood}$.
2: **Output:** Prediction $y$.
3:
4: $\{$1. Extract sample distribution$\}$
5: Compute $\mathcal{N}_x = \mathcal{N}(\mu_x, \Sigma_x)$ from $x$ using $f_v$.
6:
7: $\{$2. Group routing$\}$
8: Select $g^* \leftarrow \arg\min_{g \in G} \mathrm{KL}(t, g)$.
9:
10: $\{$3. Adapter activation and prediction$\}$
11: **if** $\mathrm{KL}(\mathcal{N}_x \| \mathcal{N}_{g^*}) < \tau_{ood}$ **then**
12:    $y \leftarrow \mathcal{M}(x; \mathcal{A}_g^*)$                ▷ In-distribution: Use corresponding IGA.
13: **else**
14:    $y \leftarrow \mathcal{M}(x)$                     ▷ OOD: Default to base model (zero-shot).
15: **end if**
16: **return** $y$

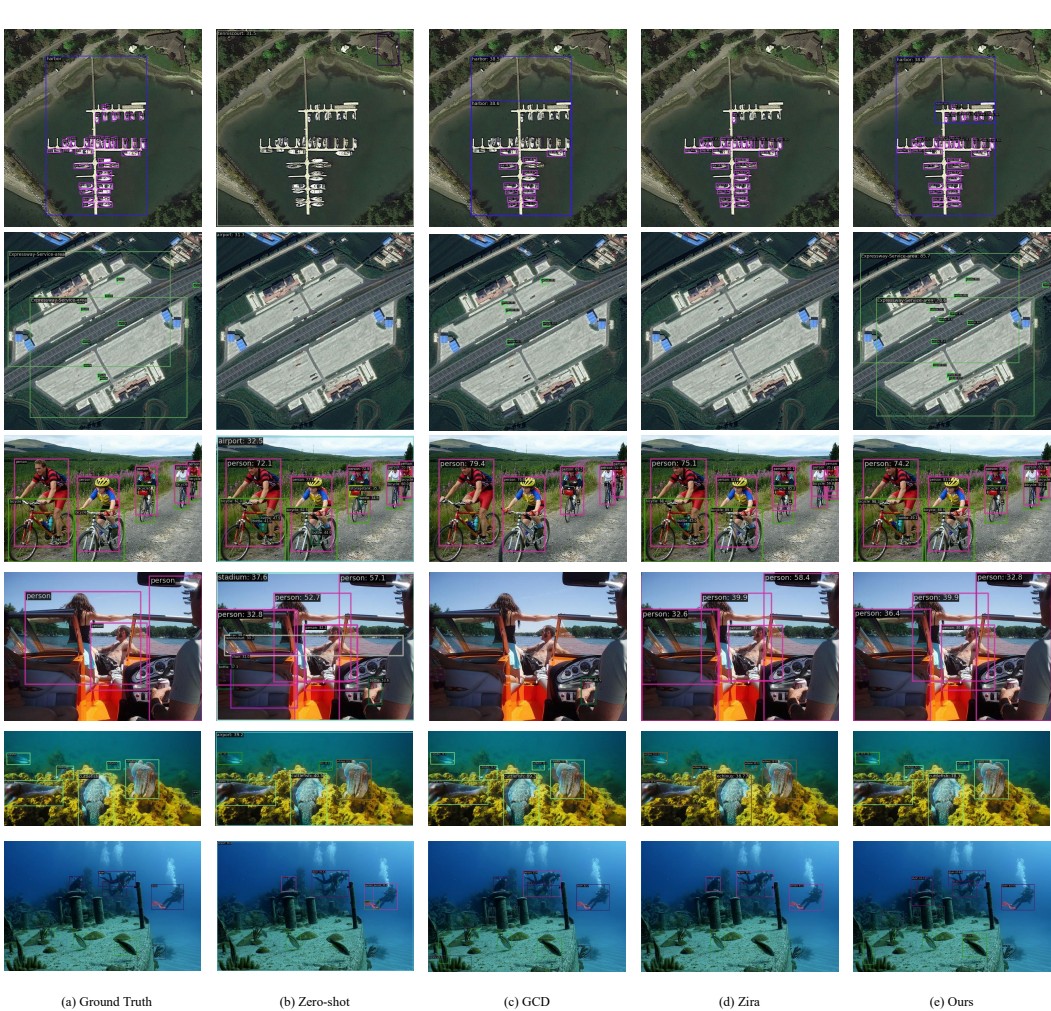

(a) Ground Truth   (b) Zero-shot   (c) GCD   (d) Zira   (e) Ours

Figure 9: Qualitative results of Zero-shot, GCD, Zira, and our method under the 0–5 (10-phase) setting. Rows 1 to 2 show samples from DIOR, 3 to 4 from Pascal VOC, and 5 to 6 from RUOD.

