# OpenReview forum: "Revisiting Incremental Object Detection with Pre-Trained Vision-Language  Models"
_ICLR.cc/2026/Conference — ICLR 2026 Conference Withdrawn Submission_

### Official Review · Reviewer_nf99 · 2025-10-25

**Soundness:** 3
**Presentation:** 3
**Contribution:** 2
**Rating:** 4
**Confidence:** 4

**Summary:**

This paper introduces Cross-Domain Incremental Object Detection (CDIOD), a new benchmark for evaluating Vision-Language Models (VLMs) in scenarios involving both novel categories and domain shifts. To address the challenge of balancing adaptivity and stability under such conditions, the authors propose D^3, a dynamic framework integrating Dynamic Task Grouping, Dynamic Adapters Assignment, and a Dynamic Training Pipeline. Experiments on CDIOD and other benchmarks show that D^3 achieves state-of-the-art performance with minimal parameter overhead, demonstrating strong adaptability and robustness.

**Strengths:**

1. Clear Architecture Illustration. The framework diagram effectively presents the overall structure and workflow of D^3, helping readers grasp the method intuitively.

2. Well-Explained Method Section. Section 4 clearly describes each component of D^3, improving readability and easing understanding of the proposed approach.

3. Comprehensive Ablation Studies. The ablation experiments are detailed and informative; for example, Figure 5 provides valuable insights and reference for future research.

**Weaknesses:**

1. The proposed method essentially follows the routing-based architectural extension paradigm, similar to earlier approaches such as L2P and MD-DETR. The overall design philosophy, introducing modular routing for incremental adaptation, has been well established in prior works, and this paper does not introduce substantial methodological innovation beyond these existing frameworks.

2. The strategy for mitigating forgetting of pre-trained knowledge is overly simplistic. The method primarily discards or deactivates previously introduced parameters, which is conceptually similar to techniques employed in most previous methods. In contrast, the design of ZiRa provides a more elegant and principled mechanism for preserving pre-trained knowledge while maintaining adaptability. The proposed solution here appears less sophisticated and lacks theoretical depth.

3. Tables 1 and 2 fail to include evaluations that directly measure the retention of pre-trained knowledge, which is crucial for validating the claimed effectiveness of the approach. Moreover, in Table 3, the performance improvement over ZiRa is marginal. This limited gain suggests that the proposed design, despite its complexity and lengthy formulation, may contribute only minor practical benefits.

4. Although the paper introduces a new benchmark termed CDIOD, its differences from existing continual learning setups are minor. Essentially, the benchmark extends conventional multi-dataset incremental learning by adding a few more incremental steps within a single dataset rather than truly capturing the dynamics of realistic, evolving domains. As a result, it does not meaningfully advance the field toward more practical or real-world incremental learning scenarios.

**Questions:**

See weakness.

---

### Official Review · Reviewer_UAo2 · 2025-10-30

**Soundness:** 3
**Presentation:** 3
**Contribution:** 2
**Rating:** 4
**Confidence:** 4

**Summary:**

This paper focuses on incremental object detection (IOD) using pre-trained visual language models (VLMs), particularly in more complex and realistic scenarios where tasks originate from multiple domains (cross-domain) rather than a single domain. The authors propose a new benchmark set, CDIOD, and a novel framework, D3, which combines dynamic task grouping, dynamic adapter assignment, and a dynamic training pipeline. Experimental results demonstrate that D3 outperforms state-of-the-art methods in handling domain transformation and incremental learning challenges.

**Strengths:**

* The discovery on performance droping when switch domains during training is interesting.
* The key idea of the proposed D3 framework, using dynamic grouping and adapter assignment, is reasonable.
* The experiments show the effectiveness of the proposed method.

**Weaknesses:**

* The motivation and methodology are mismatched. The paper's motivation emphasizes that existing work may face significant performance degradation when switching domains due to domain bias. However, the method design does not seem to have any special design for cross-domain issues, appearing no different from existing ICD methods.
* Following 1., the paper also lacks research on some existing cross-domain and PEFT related work, such as [1][2].
* As can be seen from the second row of Figure 7, the baseline model has a very strong zero-shot capability (maintaining high performance even in untrained scenarios). Therefore, the paper needs to demonstrate the performance improvement of the proposed method relative to the baseline model to prove the effectiveness of the method.

[1] Learning domain-aware detection head with prompt tuning. NeurIPS, 2023

[2] SEEN-DA: SEmantic ENtropy guided Domain-aware Attention for Domain Adaptive Object Detection. CVPR, 2025

**Questions:**

* The text in Figure 1-3 is too small and hard to read.
* I'm curious why the training is strictly based on domain switching(e.g., DIOR(2phases)→ PascalVOC(2)→RUOD(1).). I believe a good measure of cross-domain performance should randomly shuffle all classes and domains.
* Please refer to Weakness.

---

### Official Review · Reviewer_x2uv · 2025-10-31

**Soundness:** 3
**Presentation:** 2
**Contribution:** 3
**Rating:** 6
**Confidence:** 4

**Summary:**

Authors introduce Cross-Domain Incremental Object Detection (CDIOD) to evaluate incremental adaptation of VLMs under diverse domain shifts. They proposed D^3, a dynamic architecture with task grouping, adapter assignment, and training scheduling to balance adaptivity and stability. Experiments on multiple benchmarks demonstrate significant gains over IOD and IVLOD baselines.

**Strengths:**

Good motivation: Current IOD methods assume incremental phases stay in-distribution, which is not how models are deployed in practice. CDIOD is well-motivated and reflects the real-world situation.

Novel idea: The idea of grouping related tasks and assigning group adapters to let them share capacity looks sound. The routing-at-inference also looks plausible for scaling to many domains without blowing up parameters.

Good performance: The proposed method achieves strong performance with only 1.2% parameter overhead and this looks attractive.

Thorough experimental analysis: The authors distinguish “intra-domain forgetting”, “cross-domain stability” and “adaptivity” and quantify them. This seems better than just reporting mAP after each step.

**Weaknesses:**

The method depends heavily on a VLM detector backbone. It is not guaranteed that the proposed method could be applied to other types of backbones.

**Questions:**

How distribution similarity computed? It would be better to have more explanation on this.

What happens if tasks come from many small related domains and the number of adapters may explode?

---

### Official Review · Reviewer_mdU1 · 2025-11-01

**Soundness:** 3
**Presentation:** 3
**Contribution:** 3
**Rating:** 4
**Confidence:** 3

**Summary:**

This paper identifies a significant limitation in the current evaluation of Incremental Object Detection (IOD) for Vision-Language Models (VLMs). The authors argue that existing benchmarks are oversimplified, assuming incremental tasks come from a single, general domain, which does not reflect the real-world deployment of VLMs that are designed to operate across diverse domains. To address this, they propose a new benchmark, Cross-Domain Incremental Object Detection (CDIOD), which sequences tasks from three distinct domains: natural scenes (Pascal VOC), remote sensing (DIOR), and underwater (RUOD).

**Strengths:**

1. The introduction of CDIOD is a significant contribution. It moves beyond the oversimplified single-domain assumption and presents a more realistic and challenging evaluation protocol for VLM-based incremental learning. This benchmark is well-constructed, uses publicly available datasets from diverse domains, and is likely to become a valuable tool for the community.
2. The paper effectively critiques the current state of IOD research and convincingly argues why a cross-domain benchmark is necessary, especially in the context of modern, general-purpose VLMs. The analysis in Section 3.2 clearly shows the shortcomings of existing methods.
3. The method is highly parameter-efficient, adding only 1.2% trainable parameters and activating 0.4% at inference, making it very attractive for real-world applications where model size and inference cost are critical.

**Weaknesses:**

1. The authors correctly identify a key limitation: DTG's performance relies on accurate task distribution estimation, which can be unreliable with very small amounts of data (e.g., few-shot settings). While this is acknowledged, it remains a potential weakness for certain application scenarios. The robustness of DTG to very noisy distribution estimates could be explored further.
2. While comparisons to SOTA IOD methods are comprehensive, the paper could be strengthened by comparing against a broader set of general continual learning methods that also use dynamic architecture/parameter-isolation strategies, even if they are not specifically designed for object detection or VLMs. This would better situate the dynamic grouping concept within the wider field.
3. The expansion threshold τ for DTG is a critical hyperparameter. While the value of 150 is provided, an ablation study on its sensitivity and its impact on the final number of groups and performance would have been informative.

**Questions:**

See weaknesses above.

---

### Note · Authors · 2025-11-14

I have read and agree with the venue's withdrawal policy on behalf of myself and my co-authors.